# Implementation of the Helsinki Model at West Tallinn Central Hospital

**DOI:** 10.3390/medicina58091173

**Published:** 2022-08-29

**Authors:** Katrin Gross-Paju, Ulvi Thomson, Raul Adlas, Helle Jaakmees, Karin Kannel, Sandra Marii Mallene, Svetlana Mironenko, Agnes Reitsnik, Ain Vares, Sandra Ütt

**Affiliations:** 1Clinic of Neurology and Psychiatry, West-Tallinn Central Hospital, 10617 Tallinn, Estonia; 2Emed Lab, Taltech, Tallinn Technical University, Ehitajate tee 5, 12616 Tallinn, Estonia; 3Tallinn Emergency Medical Service, 13415 Tallinn, Estonia

**Keywords:** stroke, pre-hospital care, door-to-needle time, thrombolysis, service provision, outcome, quality of care

## Abstract

Ischemic stroke is defined as neurological deficit caused by brain infarction. The intravenous tissue plasminogen activator, alteplase, is an effective treatment. However, efficacy of this method is time dependent. An important step in improving outcome and increasing the number of patients receiving alteplase is the shortening of waiting times at the hospital, the so-called door-to-needle time (DNT). The comprehensive Helsinki model was proposed in 2012, which enabled the shortening of the DNT to less than 20 min. *Background and Objectives*: The aim of this study was to analyze the transferability of the suggested model to the West Tallinn Central Hospital (WTCH). *Materials and Methods*: Since the first thrombolysis in 2005, all patients are registered in the WTCH thrombolysis registry. Several steps following the Helsinki model have been implemented over the years. *Results*: The results demonstrate that the number and also the percent of thrombolysed stroke patients increased during the years, from a few thrombolysis annually, to 260 in 2021. The mean DNT dropped significantly to 33 min after the implementation of several steps, from the emergency medical services (EMS) prenotification with a phone call to the neurologists, to the setting-up of a thrombolysis team based in the stroke unit. Also, the immediate start of treatment using a computed tomography table was introduced. *Conclusions*: In conclusion, several implemented steps enabled the shortening of the DNT from 30 to 25.2 min. Short DNTs were achieved and maintained only with EMS prenotification.

## 1. Background and Objectives

The updated definition describes a stroke as a neurological deficit attributed to an acute focal injury of the central nervous system (CNS) by a vascular cause [1]. Stroke is a heterogeneous disease with mainly two subtypes, ischemic stroke (IS) and hemorrhagic stroke (HS). The definition of an ischemic stroke (IS) is based on an underlying brain infarction causing neurological deficits [1]

Intravenous thrombolysis (IVT) with alteplase for IS was introduced after a successful clinical trial (NINDS trial) in 1995 [2]. The European Medicinal Agency approved alteplase for the treatment of a stroke within 3 h after onset, on 30 November 2002 [3].

Today, this treatment is available only up to 4.5 h after the onset of IS [4]. This time frame is extended if advanced imaging demonstrates salvageable brain tissue [5]. According to the European Stroke Organization (ESO) guidelines, IVT is indicated with any stroke patient with a known time of onset within 9 h, or for patients with IS on awakening from sleep, who have a CT or MRI core/perfusion mismatch within 9 h from the midpoint of sleep [6].

The efficacy of IVT is time-dependent. Faster onset-to-treatment time was associated with reduced in-hospital mortality, reduced symptomatic intracranial hemorrhage, increased achievement of independent ambulation at discharge and increased discharge to home in 15-min increments [7]. Also, lower Rankin scores in three months was associated with shorter onset-to-treatment times [8]. Onset-to-treatment time is associated with DNT. Patients with a longer DNT (in 60-min increments) have less chance of achieving a modified Rankin Scale score of 0 to 1 at 3 months [9]. 

The analysis of several factors that influence DNT has been conducted in Helsinki, Finland [10]. The authors highlighted a comprehensive strategy of several steps, starting with collaboration with the emergency medicine services (EMS), whose dispatchers contact a stroke physician directly via mobile phone. In addition, important changes to in-hospital workflow were developed and implemented [10]. This complex approach was later called the Helsinki model [11,12].

The importance of the EMS for a faster and higher quality of stroke care is well-established. Direct training of EMS teams facilitates the better recognition of stroke [12,13,14]. Better stroke recognition was translated into slightly longer on-site times, but faster transport and decreased in-hospital management times [15]. In addition to better stroke recognition, EMS prenotification of hospitals has been recommended [10,16]. It has been shown that prenotification decreases the time for brain imaging [16,17], increases the number of patients eligible for IVT [15,17,18] and reduces mortality [18].

In some models, the EMS notifies the emergency department (ED) who then contacts the stroke team [11,19,20,21,22,23]. Alternatively, the EMS prenotify on-call neurologists in some settings, either by phone call or by SMS [10,12]. Also, direct transfer to the stroke unit, bypassing the ER, has been studied [24].

In some settings EMS prenotification is limited to stroke patients who are within the accepted time frame (2 to 4 h) for IVT, usually leaving time for in-hospital procedures [12,13,25]. 

Although there is a lot of data supporting the influence of the EMS role on the DNT and in-hospital quality of care, in some studies, the collaboration with the EMS is not described as being part of improving in-hospital DNTs [21,25,26,27,28,29,30,31]. 

The comprehensive Helsinki model comprises not only pre-hospital but also in-hospital workflow. Namely, after prenotification by the EMS, the neurologist studies statewide electronic medical records of the patient during the hospital transportation, the computed tomography (CT) and laboratory were informed, the CT was emptied, the patient was directly transferred to CT, the patient was evaluated on the CT table. The CT results were interpreted immediately by the stroke neurologist (without written report of the radiologist) and an alteplase bolus was administrated on the CT table. Later, CT was relocated to the emergency room. They also implemented suggested changes and were able to decrease DNT to 20 min [10].

In many studies, several key steps similar to those described in the Helsinki model were described. In-hospital pathways, including better collaboration of different teams and saving time through improved workflow, were implemented with a decrease in DNT [20,23,26,28,29,30,32,33]. 

The aim of the study was to describe the transferability of the key components of the Helsinki model to an Estonian stroke center over time.

## 2. Materials and Methods

### 2.1. Health Care Setting

Estonia has equal and universal coverage of emergency care for everybody, including patients without any type of health insurance. The city of Tallinn is divided among three stroke centers which offer intravenous thrombolytic therapy (IVT) with tissue plasminogen activator (tPA) alteplase. West Tallinn Central Hospital (WTCH) is serving as a stroke care facility for predefined areas in North Tallinn, Haabersti and part of Kristiine, with a small region outside of Tallinn comprising a population of about 134,500. The region is covered by EMS consisting of two independent ambulance service providers. All EMS providers work under the same guidelines. All patients within the WTCH service area with symptoms suggestive of stroke are transferred to WTCH. Patients who need a thrombectomy are transferred to the North Estonian Regional Hospital from WTCH.

### 2.2. Registry Setup

The prospective registry of all thrombolysis patients was created after the first thrombolysis on 04 April 2005. The registry has been approved by institutional authorities. As a routine observational quality registry, no patient consent for registration was required and all subsequent patients who received thrombolysis were registered. Demographic, pre-hospital and clinical data are obtained in a prospective manner.

### 2.3. Hospital Room Locator

The ER is located on the 2nd floor, accessible for EMS by ambulance. The CT and MRI are on the 3rd floor, therefore the elevator is needed to reach either the CT or MRI from the ER. The stroke unit is located on the 9th floor which is accessible only via elevators. 

### 2.4. Collaboration with EMS

Since 2006, annual meetings for 150–200 EMS staff (ambulance nurses, leaders of the ambulance teams, paramedics) have been conducted. Each year, EMS quality metrics are collected, analyzed and presented by WTCH neurologists to EMS staff during annual meetings. The presentations analyze yearly EMS onsite activities, emphasizing the importance of spending short times on-site and avoiding unnecessary procedures. Also, timely prenotification (preferably from the patient’s home) of the on-call stroke neurologist by mobile phone, documentation of the exact time of symptom onset (not 20 min ago but 12:45 for instance), blood sugar levels and used medications are analyzed from EMS charts and reported annually. Due to changes in guidelines and the decrease in absolute contraindications for IVT and thrombectomy, as well as the introduction of advanced imaging for stroke diagnosis, EMS staff are trained to manage all stroke patients with symptom onset within 12 h, as potential candidates for IVT.

### 2.5. Protocols for In-Hospital Management of Stroke

Thrombolysis protocol 2005–2009: All stroke patients were admitted to the emergency room (ER) by EMS. ER physician notified on-call neurologist. Blood tests were obtained in the ER, then CT was performed. CT report was interpreted together with the radiologist and neurologist on-call. IVT was not given before the lab test results (including INR) were available. IVT was given in the Stroke unit.

Thrombolysis protocol 2009–2013: INR point of care (POC) was implemented and IVT was given in the Stroke unit before blood test results were reported.

Thrombolysis protocol 2014–2021: EMS prenotification by mobile phone to stroke neurologist for all stroke patients within hours after onset (since November 2013), since 2016, prenotification within 12 h, and without known time of onset of stroke was implemented. Prenotification of the neurologist enables the neurologist to specify the exact time of onset, evaluate used medication and other details from EMS staff, and also to conduct a review of the statewide electronic medical records of the patient before arrival. The thrombolysis team was created, based on the stroke unit members (neurologist, stroke unit nurse). After the prenotification call to the neurologist by the EMS, the thrombolysis team meets the EMS in the ER, a quick neurological exam is performed in the ER and IVT bolus is given immediately, while the patient is on the CT table. 

Since 2016, emergency magnetic resonance imaging (MRI) for stroke without known time of onset was implemented due to the accessibility of this visualization modality. 

An overview of the standard protocol since 2014 is depicted in Figure 1.

### 2.6. Indications for IVT

Indications for IVT have changed over time. Since 2005, IVT was used if stroke occurred within 3 h after onset. Since 2008, the time window was extended to 4.5 h after onset. Since 2016, IVT was used if an MRI flair/diffusion weighted imaging mismatch was demonstrated in patients with a wake-up stroke and unknown time of onset.

### 2.7. Thrombectomy

Thrombectomy is performed at the regional hospital (6 km away, driving time 13 min) where patients are transferred to from WTCH. All patients with a planned thrombectomy without contraindications to thrombolysis receive thrombolysis prior to transfer. These patients are included in the analysis.

### 2.8. Statistical Analysis

The statistical analysis was performed with MS Excel (version 2016, Microsoft, Washington, DC, USA) software. For descriptive statistics, mean and SD values are presented. For comparisons between groups, a *t*-test was applied.

### 2.9. Ethics Approval

The registry has been approved by institutional authorities. As this is a routine observational quality registry, no patient consent for registration is required.

## 3. Results

From 2005 to 2021, 8699 strokes have been treated at WTCH. Thrombolysis was performed in 1698 patients. The number of strokes and thrombolysis from 2005 to 2021 is depicted in Figure 2 and Figure 3.

### 3.1. Pre-Hospital Care of Thrombolysed Patients

Strokes comprised only 1.28% of Tallinn EMS calls in 2021. Therefore, neurologists are involved in the annual analysis and feedback of the quality of EMS services of pre-hospital stroke care.

Quality metrics of the EMS were analyzed for four periods: 2009–2011, 2016, 2020, 2021.

The changes are depicted in Table 1. The number of neurologist prenotifications increased to 88%, and the number of performed ECGs decreased to 22%. The number of documented blood sugar values was 84% during 2021. The documentation of used medication was 98% in 2021.

### 3.2. In Hospital Door-to-Needle Times

Door-to-needle times from 2005 to 2021 are depicted in Figure 4.

The mean DTN time was 85 min (93…84) from 2005 to 2008, with tendencies towards somewhat decreasing numbers. The DNT was unchanged after the new IVT standard procedure was introduced without lab results; INR POC was implemented in 2013. From November 2013, the EMS and Tallinn stroke centers agreed the prenotification of neurologists. At WTCH, the procedure was also changed to include the reviewing electronic health records before the patient’s arrival, the thrombolysis team meeting the EMS at ER, neurological exam on CT table, thrombolysis started on CT table (Figure 1). These amendments quickly enabled a decrease in DNTs to a mean of 33 min, which was statistically significant from 2013. By 2021, the proportion of patients thrombolysed in under 20 min increased to 37–42% (Figure 5). Since 2016, advanced imaging was used, according to ESO guidelines [6]. Although the mean times increased to 40 min, the DNT for CT-based thrombolysis remained the same (Figure 4). Also, the number of patients thrombolysed in under 20 and 30 min is stable (Figure 5).

We also analysed the influence of the EMS prenotification on DNTs in 2020 and in 2021, for strokes from the Tallinn city service area, excluding referrals from outside hospitals. 

In 2020, the mean DNT for 141 patients whose thrombolysis started in the CT, was 25.5 min, compared to 22 patients who arrived without prenotification to CT with a mean DNT of 60.4 min (*p* = 0.003). In 2021, the mean DNT for 192 patients whose thrombolysis started in the CT was 25.4 min, compared to 14 patients who arrived without prenotification to CT with a mean DNT of 89.1 min (*p* = 0.016). 

MRI has been used for determining thrombolysis since 2016. Thrombolysis was used after adavanced imaging in 21% (2016), 21% (2017), 18% (2018), 19% (2019), 14% (2020) and 14% (2021) of thrombolysed patients. 

In 2020, 29 patients were thrombolysed after advanced imaging with MRI. The mean DNT for 23 patients with prenotification was 70.7 min to MRI, compared to six patients without prenotification where the mean DNT to MRI was 157.9 min.

In 2021, 36 patients were thrombolysed after advanced imaging with MRI. The mean DNT for 26 patients with prenotification was 72.2 min to MRI compared to 10 patients without prenotification where the mean DNT to MRI was 156.6 min.

Since 2005, the number of thrombolysed stroke patients increased from a few patients to 260 thrombolysis in total, and to 246 in 2021, when stroke mimics are excluded. 

## 4. Discussion

The definition of ischemic stroke (IS) is based on an underlying brain infarction causing neurological deficits [1]. Since 1995, after the NINDS trial, IVT was established as an effective treatment [2], and it has been available in Europe since 2002 [3]. Efficacy of IVT is time-dependent [7,8,9] and therefore a short time from onset to IVT is important. Patients with longer DNTs have less chance of a good outcome [9]. Several studies have addressed different steps to reduce DNTs. Some approaches to decrease DNTs address collaboration with the EMS [10,11,12,13,14,15,16,17,18,19,22,23,24,25], while others developed and implemented in-hospital workflow [21,25,26,27,28,29,30,31].

In order to improve the quality of stroke care at WTCH, recommendations from a seminal paper published in 2012 by Meretoja [10] were analyzed and most of them implemented.

### 4.1. Collaboration with EMS

The direct training of EMS staff is crucial for the immediate recognition of stroke [10,12,13,14,15]. In our model, we introduced broad-based annual meetings for EMS staff to improve stroke recognition, quality and speed of onsite activities and the prenotification of neurologists. 

Several studies have identified the group of stroke patients who need prenotification differently. In older studies, when advanced imaging was not part of clinical routine, specific onset times were defined for prenotification [12,13]. In our model, we implemented prenotification of all stroke patients with unknown onset, or onset within 12 h, taking into account the advanced imaging approach which allows for the increase in the number of patients receiving IVT [6]. According to the Helsinki model [10] and our experience, EMS prenotification of the neurologist by phone call is preferred, since during the short conversation, details about stroke onset and the patient’s identity number, needed to access electronic medical records, are delivered directly and in a timely manner. In line with the Helsinki model, it is important to access comprehensive electronic patient health care records during patient transport [10], and at our center, it saves time compared to paper-based documentation and a phone call to the general practitioner [11].

Therefore, SMS [11,12] or ER prenotification [11,18,19,22,23] may be less time-saving. After the phone call, the on-call neurologist informs the stroke unit nurse, a member of the thrombolysis team. At our center, immediately after the implementation of prenotification, only 29% thrombolysed patients arrived with prenotification from the EMS compared to 88% in 2021 (Table 1). 

The analysis of the EMS onsite quality metrics is important in decreasing the time onsite and for improving the quality of care. For instance, performing ECG onsite is not necessary according to EMS guidelines [34]. Also, we have demonstrated that performing ECG results in an EMS response time of 26.8 min compared to an onsite time of 18 min without this procedure [35]. The proportion of performed ECG procedures in our study decreased from 51% in 2016, to 22% in 2021. On the other hand, blood sugar values were documented in only 67% (2005–2011) and in 84% in 2021. The used medications are well documented in 98% of thrombolysed patients in 2021 (Table 1). 

### 4.2. Changes in Door-to-Needle Times

Our experience demonstrates that in order to decrease DNTs significantly, several simultaneous steps are necessary. After the implementation of prenotification, electronic health records review before patients’ arrival, stroke team taking over the stroke patient at the ER door, immediate CT and administration of IVT bolus on CT table, the average DNT decreased statistically significantly (from 40 to 33 min) (Figure 4).

It was found that 37% of all thrombolysed patients had a DNT of less than 20 min, and 64%, less than 30 min. It was also found that 86% of all thrombolysed patients received IVT within 1 h after admission (Figure 5).

Interestingly, even with all in-hospital procedures in place, the influence of hospital prenotification still had a major influence on DNT. Immediately after hospital prenotification was implemented, there were significant differences between DNTs with and without prenotification [35]. According to our present data in 2021, the mean DNT for 190 patients whose thrombolysis started in the CT, was 255 min, which was unchanged from 2016 [35], compared to a mean DNT of 89.2 min for those arriving without prenotification. EMS prenotification also significantly influenced the DNT for those patients who received thrombolysis after MRI was performed. DNTs for MRI are still unacceptably high, both with and without EMS prenotification, and needs to be improved.

One important step to decrease DNTs would be the relocation of the CT to the ER. Interestingly, for a short period when the ER CT was repaired and the other CT had to be used, it did not change the DNTs significantly in Helsinki [10]. Due to the CT and MRI location being one floor up from the ER, which makes the elevator use mandatory, we suggest that for our center, the location of the CT may increase DNTs.

In addition, to shorten DNTs, we also observed a significant increase in the number of thrombolysed patients. In 2005–2008, we thrombolysed 7–17 patients annually. In 2021, 260 thrombolysis were performed, including 246 strokes and 14 stroke mimics (5%). Advanced imaging is also used in a proportion of patients without a certain time of onset. Since 2018, 14% to 21% of thrombolysis were performed after salvageable tissue was demonstrated by MRI.

## 5. Conclusions

In conclusion, our study demonstrates that the key steps developed and implemented in Helsinki for decreasing DNTs are transferrable to other centers, and short DNTs are achievable. However, even though most steps were implemented at WTCH, the DNTs are still higher, 25.5 min vs. less than 20 min in Helsinski [10]. Although the location of the CT outside of the ER did not result in longer DNTs in Helsinki, in our setting, with the CT on the 3rd floor and the need to use elevators, it further limits the shortening of DNTs.

In conclusion, the Helsinki model is well designed, transferrable and represents a working model for decreasing DNT for stroke.

## Figures and Tables

**Figure 1 medicina-58-01173-f001:**
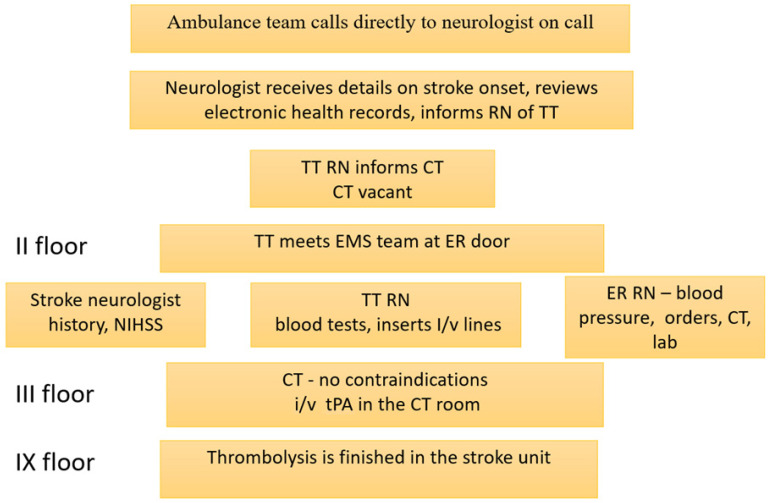
Thrombolysis protocol at WTCH 2014–2021. RN—registered nurse, TT—thrombolysis team, ER—emergency room, CT—computed tomography, EMS—emergency medical team, ambulance, tPA—tissue plasminogen activator, i/v—intravenous, NIHSS—National Institute of Stroke Scale.

**Figure 2 medicina-58-01173-f002:**
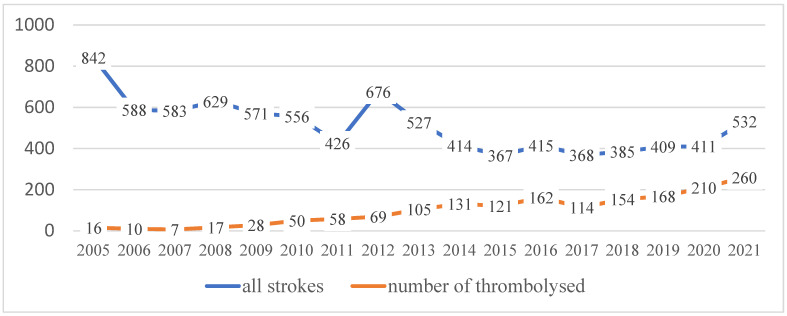
The number of treated stroke patients and number of thrombolysis (including stroke mimics) since 2005.

**Figure 3 medicina-58-01173-f003:**
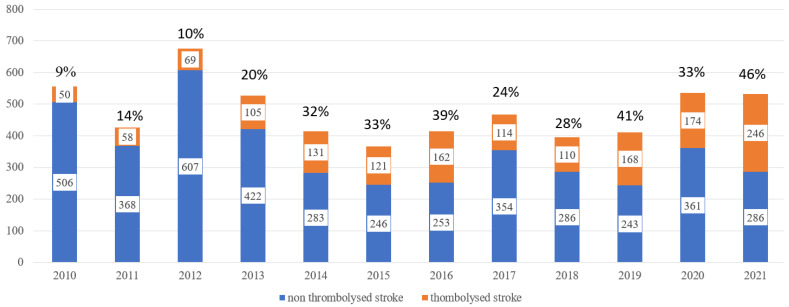
All stroke and thrombolysed stroke patients and percent of thrombolysed patients since 2010. Thrombolysed stroke mimics since 2018 are excluded.

**Figure 4 medicina-58-01173-f004:**
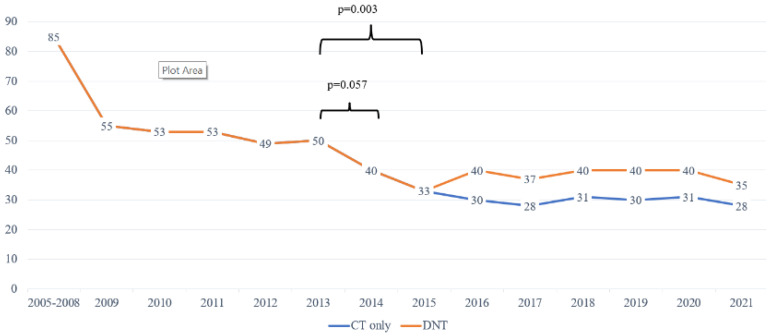
The mean door-to-needle times (DNT) at WTCH over the years.

**Figure 5 medicina-58-01173-f005:**
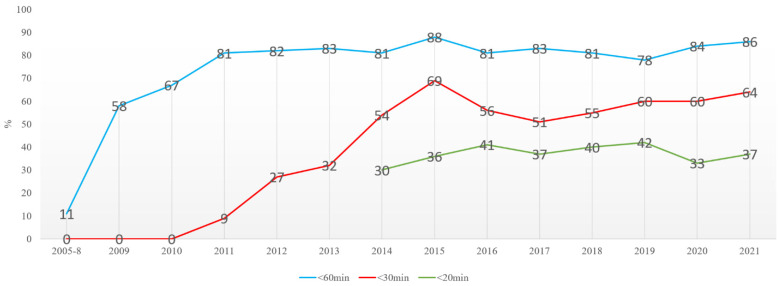
The number of patients thrombolysed under 20 and 30 min.

**Table 1 medicina-58-01173-t001:** Onsite activities of EMS from 2009 to 2021.

	2009–2011	2016	2020	2021
Pre-notification to neurologist	-	29%	83%	88%
ECG	51%	41%	20%	22%
Blood sugar	67%	93%	85%	84%
Documentation medications	69%	87%	97%	98%

ECG—electrocardiography, EMS—emergency medical services.

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
