# Peer review of "Implementation of the Helsinki Model at West Tallinn Central Hospital"

_medicina, 2022, doi:10.3390/medicina58091173_

Round 1

Reviewer 1 Report

There are the following important issues that should be able to be understood
in reading the article:
1/ The Introduction is too long and it seems more an historical description about
the thrombolysis.
2/ Given the article is a description, there is a lack of data about the epidemiology
of stroke (ex. incidence, severity, average people affected profile, demographic aging, etc...) so that the reader can better understand the context of the description.
3/ A description of the before/after differences that have made it possible to obtain these
results is not included. The authors have written the process, but not how it was done.
It does not provide data about what changes were made to achieve them, nor a detailed
description about results in disability avoided, data comparing the population served,
structural changes (ex. neurologist place, who decides the call, who evaluated NIHSS score,
stroke severity.....), and/or explain in Discussion the evolution (line 167) of 24 to 46% over
the last 7 years in the proportion of thrombolysed stroke patients.

4/
The stroke neurologist is not located in the hospital? 5/ Who performs the evaluation of the clinical severity of stroke before pre-notification?
What criteria do you use to perform thrombolysis?
6/ Who makes the pre-notification? And the phone-call? 7/ Just the pre-notification (29% to 88%) to stroke neurologist improved (Table 1).
Explains only this the improvement in the DNT?
8/ The beginning of the Discussion is a repetition of previous text. It should be rewritten. 9/ About the analysis of quality metrics of EMS in the proportion of EC (51% to 22%),
I suggest to revise the incidence of stroke related to unknown atrial fibrillation and reconsider
whether this progression is a positive quality indicator.
10/ In this long time, was introduced the thrombectomy? It has not been commented. 11/ / The reference (line 39) should be rewritten and the references (n.d. ex. lines 53, 68)
removed because there are enough evidence with references to be included.
Eventually, the authors concluded “our experience demonstrates that in order to decrease
significantly DNT several simultaneous steps are necessary” (line 263), but what steps
did they do? since 2005-2021 (simultaneous?), and what was the evolution of health outcomes
(incidence, disability, severity, etc...) before/currently?

Author Response

Dear reviewer,

thank you very much for your valuable comments.I have updated methods and discussion according to your comments.

However, as the paper was a description of implementation of technical steps to decrease door-to-needle time according to suggestions from paper by Meretoja et al. Therefore the aim was not to evaluate stroke and outcomes specifically. The importance of short onset to treatment times and short DNT have been demonstrated in earlier studies. Current description of multiple changes over time to not allow to do this type of evaluation correctly, according to our opinion.

But this is true that this type of evaluation is very important to understand if  we are moving in the right direction with our chnages.

Thank you very much for your effort,

Katrin Gross-Paju

Reviewer 2 Report

Comments and Suggestions for Authors: 

The abstract must be adequate to 250 words. The content of the introduction section should be reduced and the material and methods section expanded.

In the introduction and discussion section, the authors should discuss the existence of factors that influence prehospital delay. As an example, the following bibliographical references are indicated:

Factors associated with a rapid call for assistance for patients with ischemic stroke - EMERGENCIAS - 2020

Factors associated with the activation of emergency medical services in patients with acute stroke: a prospective study - EMERGENCIAS 2019

The authors should redo the material and methods section, adapting to the following scheme: type of design, population, study sample,  procedure, ethical considerations (favorable report from the institution's ethics committee), study variables and method of data collection, statistical analysis.

The authors should consider performing new inferential statistical analyses, and not just limit themselves to describing the results obtained.

The authors should expand the results obtained

The discussion section should start with a brief summary of the results obtained. This section hardly compares with the results obtained in other studies. This section should be expanded. The strengths of the study, its strengths and future lines of research are missing.

The bibliographical references must be adapted to the indications of the journal.

The authors must be careful with the statements they make throughout the study, since given the design of the study, the existence of causality cannot be determined.

Author Response

Dear Reviewer,

Thank you very much for your valuable comments. I rewrote the abstract and introduced other changes that you have proposed.

Thank you very much for your comments,

Katrin Gross-Paju

Round 2

Reviewer 1 Report

To Authors,

related your comments: 

1/ I agree about "as the paper was a description of implementation of technical steps to decrease door-to-needle time according to suggestions from paper by Meretoja et al"

2/ But not about:

2.1. "the aim was not to evaluate stroke and outcomes specifically", because you included as keywords "outcome" and "quality care"

2.2. "The importance of short onset to treatment times and short DNT have been demonstrated in earlier studies" but it does not guarantee without exception a better clinical prognosis. Given the stroke is a complex episode, their prognosis is not just depending of a schudeled DNT (what is a great achivement), but because there are a lot of other conditions that can affect the final result of your approach. It could be better understood if you had described, as previously indicated,  basal characteristics of attended people, stroke severity, differences between people treated with/without thrombolysis, mortlity, thrombectomy? and quality metrics.

2.3. I'm not totally agree with "Current description of multiple changes over time to not allow to do this type of evaluation correctly, according to our opinion", because it would be very interesting to be able to compare possible associations between the evolution of DNT-times amd characteristics, results and prognosis of stroke. since they could indicate the good direction of the changes made. You should explain this point, at least, in the Discussion. 

Author Response

Dear Reviewer,

Thank you for your valuable comments. I included your suggestion about changes in stroke patients cohorts in the discussion and removed from key words quality and outcome. Thank you for indicating this discrepancy.

Unfortunately, the analysis of the patient characteristics was not planned to be part of the study.

Thank you very much for helping to improve the paper significantly

Katrin